# Common and Rare *PCSK9* Variants Associated with Low-Density Lipoprotein Cholesterol Levels and the Risk of Diabetes Mellitus: A Mendelian Randomization Study

**DOI:** 10.3390/ijms231810418

**Published:** 2022-09-08

**Authors:** Lung-An Hsu, Ming-Sheng Teng, Semon Wu, Hsin-Hua Chou, Yu-Lin Ko

**Affiliations:** 1The First Cardiovascular Division, Department of Internal Medicine, Chang Gung Memorial Hospital and Chang Gung University College of Medicine, Taoyuan 33305, Taiwan; 2Department of Research, Taipei Tzu Chi Hospital, Buddhist Tzu Chi Medical Foundation, New Taipei City 23142, Taiwan; 3Department of Life Science, Chinese Culture University, Taipei City 11114, Taiwan; 4Cardiovascular Center, Division of Cardiology, Department of Internal Medicine, Taipei Tzu Chi Hospital, Buddhist Tzu Chi Medical Foundation, New Taipei City 23142, Taiwan; 5School of Medicine, Tzu Chi University, Hualien City 97004, Taiwan

**Keywords:** *PCSK9* gene, Mendelian randomization, low-density lipoprotein cholesterol level, Taiwan biobank, diabetes mellitus

## Abstract

PCSK9 is a candidate locus for low-density lipoprotein cholesterol (LDL-C) levels. The cause–effect relationship between LDL-C levels and diabetes mellitus (DM) has been suggested to be mechanism-specific. To identify the role of *PCSK9* and genome-wide association study (GWAS)-significant variants in LDL-C levels and the risk of DM by using Mendelian randomization (MR) analysis, a total of 75,441 Taiwan Biobank (TWB) participants was enrolled for a GWAS to determine common and rare *PCSK9* variants and their associations with LDL-C levels. MR studies were also conducted to determine the association of *PCSK9* variants and LDL-C GWAS-associated variants with DM. A regional plot association study with conditional analysis of the *PCSK9* locus revealed that *PCSK9* rs10788994, rs557211, rs565436, and rs505151 exhibited genome-wide significant associations with serum LDL-C levels. Imputation data revealed that three rare nonsynonymous mutations—namely, rs151193009, rs768846693, and rs757143429—exhibited genome-wide significant association with LDL-C levels. A stepwise regression analysis indicated that seven variants exhibited independent associations with LDL-C levels. On the basis of two-stage least squares regression (2SLS), MR analyses conducted using weighted genetic risk scores (WGRSs) of seven *PCSK9* variants or WGRSs of 41 LDL-C GWAS-significant variants revealed significant association with prevalent DM (*p* = 0.0098 and 5.02 × 10^−7^, respectively), which became nonsignificant after adjustment for LDL-C levels. A sensitivity analysis indicated no violation of the exclusion restriction assumption regarding the influence of LDL-C-level-determining genotypes on the risk of DM. Common and rare *PCSK9* variants are independently associated with LDL-C levels in the Taiwanese population. The results of MR analyses executed using genetic instruments based on WGRSs derived from *PCSK9* variants or LDL-C GWAS-associated variants demonstrate an inverse association between LDL-C levels and DM.

## 1. Introduction

Elevated low-density lipoprotein cholesterol (LDL-C) levels constitute a major risk factor for atherosclerotic cardiovascular disease, as well as cardiovascular and total mortality [1,2,3]. A reduction in LDL-C has been consistently associated with a decrease in major vascular events, with similar effectiveness in men and women, high- and low-risk subgroups, ethnically diverse populations, patients using statin or non-statin drugs, and children with familial hypercholesterolemia [2,3,4,5,6,7,8]. Mendelian randomization (MR) studies have demonstrated a cause–effect relationship between LDL-C levels and cardiovascular disease and mortality [9,10,11,12]. However, clinical trials and meta-analyses have revealed that statins increase the incidence of new-onset diabetes mellitus (DM) in a dose-dependent manner, especially in prediabetes [13,14,15,16]. Furthermore, MR studies using variants in *HMGCR*, *NPC1L1*, and *PCSK9* have reported that decreased LDL-C levels were associated with an increased incidence of new-onset DM [9,10,17,18,19,20], but this finding was not supported by other studies [11,21]. Whether PCSK9 inhibitors increase the risk of diabetes is also still under debate [22].

Proprotein convertase subtilisin/kexin type 9 (PCSK9) is a multifaceted serine protease that plays a critical role in the regulation of hepatic LDL receptor (LDLR) function, and is involved in atherothrombosis and cardiovascular biology in addition to LDL regulation [23,24]. *PCSK9* is primarily produced in the liver. Nevertheless, it is expressed in other organs, such as the kidneys, pancreas, and brain [25]; in various cell types involved in the development of atherosclerosis, such as endothelial cells, macrophages, and smooth muscle cells; and in human atherosclerotic plaques [26]. Human *PCSK9* is located on the chromosome 1p32.3, which comprises 12 exons that encode a 692-amino-acid proteinase [26]. More than 850 *PCSK9* mutations have been reported, and studies have demonstrated that the more common loss-of-function (LOF) mutations are associated with hypocholesterolemia and protection against coronary heart disease, but that the disease-related dominant gain-of-function (GOF) mutations cause familial hypercholesterolemia and coronary heart disease [23,24]. Both GOF and LOF mutations may occur as common or rare alternative allele frequencies, and through different mechanisms [23,24,27]. Research has also commonly observed differences in genetic variants in *PCSK9* between Caucasian and East Asian populations, demonstrating the ethnic heterogeneity of such variants [28,29,30,31,32,33,34,35,36]. In contrast, noncoding *PCSK9* sequence variants are rarely reported [10,37,38,39]. Therefore, searching for ethnicity-specific variants is crucial for precision medicine for each population. The Taiwan Biobank (TWB) population-based cohort study enrolled more than 100,000 volunteers aged 30–70 years with no history of cancer [40,41]. The objectives of the present study were to apply a regional plot association analysis and an MR analysis to identify the common and rare *PCSK9* variants and other candidate variants associated with LDL-C in participants selected from the TWB cohort, and to test the cause–effect relationship between LDL-C levels and DM.

## 2. Results

### 2.1. Regional Plot Association Analysis for the PCSK9 Region

The selection process of the TWB participants is shown in Figure 1. Baseline characteristics of the study population are shown in Appendix A. Using genome-wide association study (GWAS) data obtained from 75,441 TWB participants, we performed a regional plot association study with conditional analysis to determine the association of LDL-C levels with 330 single-nucleotide variants (SNVs) located between 55.40 and 55.65 Mb on chromosome 1p32.3 within the *PCSK9* region. A total of 66 SNVs exceeded the genome-wide significance threshold (*p* < 5 × 10^−8^), with the lead SNV being rs10788994 (*p* = 2.71 × 10^−14^; Figure 2A). We also performed stepwise analysis to clarify whether the associations between the other *PCSK9* SNVs were independent of the lead SNV. After adjustment for the rs10788994 genotypes, we observed that rs565436 in the regional plot near the *PCSK9* locus was still significantly associated with LDL-C levels (*p* = 3.28 × 10^−9^, Figure 2B). After further adjustment for both rs10788994 and rs565436 genotypes, we observed that rs505151 and rs12067569 were still associated with LDL-C levels (*p* = 1.83 × 10^−7^; Figure 2C). Because of the complete LD between the rs505151 and rs12067569 genotypes (Figure 3B), we included the rs505151 genotype in our subsequent analyses. We continuously adjusted for the rs10788994, rs565436, and rs505151 genotypes; however, none of the SNVs in the regional plot near the *PCSK9* locus exhibited genome-wide significant associations with LDL-C levels (Figure 2D). Twelve SNVs located within the *PCSK9* region reached genome-wide significance in initial regional association analysis, and were selected for linkage disequilibrium (LD) analysis. We observed that three SNVs—namely, rs10788994, rs557211, and rs565436—were strongly associated with LDL-C levels in low LD (pairwise r^2^ < 0.3), and all three SNVs together with the rs505151 variant were subsequently used for *PCSK9*-weighted genetic risk scores (WGRSs) and MR analysis (Figure 3, Appendix A).

### 2.2. Association between Rare PCSK9 Nonsynonymous Exonic Mutations and LDL-C Levels

From the pre-quality-control (QC) imputation data, we found eight rare (minor allele frequency < 0.01) exonic nonsynonymous mutations of *PCSK9*, and analyzed the association of these mutations with LDL-C levels (Appendix A). All eight mutations exhibited a weak LD with the other genotypes (r^2^ < 0.01) (Figure 3C). After adjustment for age, sex, body mass index (BMI), and current smoking status, we observed that the rare alleles of three *PCSK9* variants—namely, rs151193009, rs768846693, and rs757143429—exhibited a genome-wide significant association with lower LDL-C levels (Table 1).

### 2.3. Stepwise Linear Regression Analysis

We applied additive models to execute a stepwise linear regression analysis using age, sex, BMI, current smoking status, and seven *PCSK9* variants. The analysis revealed that the rs10788994, rs151193009, rs557211, rs768846693, rs757143429, rs565436, and rs505151 genotypes contributed to 0.07%, 0.13%, 0.01%, 0.06%, 0.04%, 0.05%, and 0.02% of the variation in total cholesterol levels, respectively, and to 0.09%, 0.14%, 0.01%, 0.11%, 0.05%, 0.05%, and 0.03% of the variation in LDL-C levels, respectively (Table 2).

### 2.4. Association between LDL-C Levels and DM Status

The associations between LDL-C levels and DM status are presented in Appendix A. After adjustment for age, sex, BMI, and current smoking status, we observed that circulating LDL-C levels were negatively associated with DM status (OR = 0.99 (95% CI: 0.99, 0.99) per mg/dL unit of LDL-C, or 0.78 (95% CI: 0.75, 0.81) per mmol/L unit of LDL-C; *p* = 6.76 × 10^−64^) (Appendix A).

### 2.5. GWAS Analysis for LDL-C Levels

After fitting our linear regression model for genotype trend effects and adjusting for age, sex, BMI, and current smoking status, we observed that *APOE* rs7412 located on chromosome 19q13.32 had the highest −log_10_
*p* value for LDL-C levels (*p* < 10^−307^), and that a total of 47 lead SNVs for each peak (the SNV with the lowest *p*-value for each locus) exhibited genome-wide significant associations (Appendix A). We also assessed the associations of individual LDL-C-level-determining SNVs with DM after further adjustment for LDL-C levels; the results indicated that six SNVs–namely, *APOE* rs7412, *PMFBP1* rs3852789, *TMEM8A* rs375498857, *SLC10A1* rs2296651, *TRIM5* rs16934050, and *MACO1* rs61775184—were significantly associated with DM (all *p* < 0.01). These six SNVs were excluded from the calculation of WGRSs for the analysis of DM status. Thus, 41 SNVs were used as LDL-C GWAS-associated variants for WGRS calculation, and the resulting association with LDL-C levels increased significantly (*p* < 10^−307^; Table 3).

### 2.6. Association between LDL-C-Level-Associated PCSK9 Genotypes and Clinical and Laboratory Parameters

Over 70,000 volunteers participated in the genotype–phenotype association analysis. We analyzed the association of the seven aforementioned LDL-C-level-associated *PCSK9* genotypes with various clinical phenotypes and laboratory parameters. Using the Bonferroni correction and our additive model, and after adjusting for age, sex, BMI, and current smoking status, we observed no significant associations (*p* < 0.001) between the *PCSK9* genotypes and phenotypes, apart from LDL-C levels (Appendix A).

### 2.7. MR Analysis along with 2SLS IV Regression for Determining the Association of Genetic Determinants of LDL-C Levels with DM Status

We performed two-stage least squares regression (2SLS) instrumental variable (IV) analysis to determine the direction and causality of the association between LDL-C levels and DM status. In this regression, we used the seven aforementioned *PCSK9* variants associated with LDL-C levels to derive weighted genetic risk scores (WGRSs), and the scores revealed that LDL-C levels were significantly associated with a lower risk of DM (*p* = 0.0079; Table 3). The association of *PCSK9*-WGRS with DM status remained significant after adjustment for multiple parameters associated with LDL-C levels (*p* = 0.0098), but the association became nonsignificant after adjustment for LDL-C levels (*p* = 0.1526). From the GWAS for LDL-C levels, we selected 41 independent SNVs as IVs. All SNVs were significantly associated with LDL-C levels at the genome-wide significance level (Appendix A). We further tested the associations of the WGRSs of these 41 LDL-C-determining variants with DM status. In the 2SLS IV analysis, our data revealed significant associations between the WGRSs of the 41 gene variants and DM status (*p* = 0.0046), which remained significant even after further adjustment for multiple parameters associated with LDL-C levels (*p* = 5.02 × 10^−^7); nevertheless, the associations became nonsignificant after adjustment for LDL-C levels (*p* = 0.7606). The *F* statistics derived for the instruments ranged from 716 to 729 for LDL-C-level-determining genotypes, demonstrating a low risk of weak instrument bias (Appendix A).

### 2.8. Scatter Plots and Test for Heterogeneity

As shown in Appendix A, scatterplots of the effect sizes of the associations between the LDL-C-level-determining alleles and LDL-C levels versus the effect sizes of the association between the LDL-C-level-determining alleles and DM status in the MR analysis revealed that independent genetic variants in different gene regions were concordantly associated with the outcomes, supporting a causal relationship between LDL-C levels and DM (Appendix A). Through visual inspection and by using Cochran’s Q test, we found no evidence of heterogeneity for causal estimates in each genetic variant during the various sensitivity analyses (Appendix A).

### 2.9. Sensitivity Analysis for Causal Inference from Standard Mendelian Randomization with Multiple Genetic Variants Determining LDL-C Levels

The causal effects calculated using sensitivity analyses in the standard MR study were almost identical to the 2SLS estimates (Appendix A). The symmetric funnel plots confirmed the absence of a directional pleiotropy effect for each set of IV estimates and the IV strength (Appendix A). Moreover, our data indicated that the estimates of the effect size in standard MR analysis in the inverse-variance-weighted (IVW), median-based, and MR–Egger regression (slope) methods were all consistent with the estimates for the effect of the LDL-C-level-determining allele on DM in the 2SLS method (Appendix A). The intercept of all Egger regression analyses was close to zero. These results indicate the presence of balanced pleiotropy in our standard MR analysis.

## 3. Discussion

This study analyzed *PCSK9* variants associated with LDL-C levels in Taiwanese participants. Seven *PCSK9* coding and noncoding variants were significantly associated with both total and LDL-C levels, contributing 0.38% and 0.49% of the variations, respectively. Our genotype–phenotype association analysis revealed that these variants had no other pleiotropic effects. This study is the first to report a rare *PCSK9* coding-sequence variant—namely, rs768846693 pS249R—as a possible LOF mutation. We performed an MR analysis using 2SLS regression and a *PCSK9* WRGS genetic instrument, and the analysis results indicated that genetically determined LDL-C levels were inversely associated with the prevalence of DM. These results provide further evidence that *PCSK9* variants play a crucial role in determining cardiometabolic outcomes in the Taiwanese population.

### 3.1. Ethnic Heterogeneity of PCSK9 Gain-of-Function and Loss-of-Function Mutations

As of April 2022, more than 850 unique *PCSK9* variants have been listed in the ClinVar database (https://www.ncbi.nlm.nih.gov/clinvar/?term=PCSK9[gene] (accessed on 24 May 2022)), of which 643 are from exonic, 128 are from intronic, and 67 are from either 5′ or 3′ untranslated regions. Of the exonic variants, 384 are missense, 18 are nonsense, 22 are frame shifts, and 219 are synonymous mutations. A clinical classification indicated the existence of 34 pathogenic or likely pathogenic variants and 99 conflicting interpretations of pathogenic variants. In the present study, we observed that three rare *PCSK9* coding variants—namely, rs151193009 pR93C, rs768846693 pS249R, and rs757143429 pR434W—were LOF mutations, and were significantly associated with lower LDL-C levels. In the ClinVar database, pR93C and pR434W were classified as variants of conflicting interpretations, and S249R was classified as a variant of uncertain significance. This study is the first to report that pS249R is an LOF mutation and is associated with low LDL-C levels. Although the three *PCSK9* coding variants (rs151193009 pR93C, rs768846693 pS249R, and rs757143429 pR434W) are rare mutations (minor allele frequencies = 0.004–0.027%), their LDL-C-lowering effect per allele was noted to be significant, ranging from 13 to 29 mg/dL. A recent study reported that the *PCSK9* R93C variant was associated with a 60% lower risk of premature myocardial infarction in the Chinese Han population [42]. The ASPirin in Reducing Events in the Elderly trial also demonstrated that *PCSK9* pR434W is a member of lipid-lowering *PCSK9* and *APOB* genetic variants that are carried by healthy older individuals (aged ≥ 70 years) and contribute to coronary-heart-disease-free survival [43]. However, *PCSK9* pS249R was first identified as an LOF mutation, but its clinical implications warrant further investigation. In contrast, we found that the common variant rs505151 (E670G) was a GOF mutation and was significantly associated with higher LDL-C levels, which is consistent with previous reports [24,29,35,44].

### 3.2. Role of Noncoding PCSK9 Variants

Previous studies have focused mostly on the role of exonic *PCSK9* mutations in LDL-C levels. Our data reveal that variants in the promoter region, intronic region, and 3′ untranslated region may play a crucial role in and be independently associated with LDL-C levels. Our study revealed the *PCSK9* rs10788994 variant, located in the promoter region of *PCSK9*, as the lead SNV for LDL-C levels. However, a case–control study on premature myocardial infarction in the Italian population reported that this SNV was not associated with LDL-C levels [38]. Our data also revealed that the *PCSK9* intron variants rs557211 and rs565436 were independently associated with LDL-C levels. The rs565436 variant was reported to be associated with familial hypercholesterolemia in Malaysia [45], but the association between rs557211 and LDL-C levels has not yet been reported. Additional functional studies are necessary to elucidate the mechanism underlying the influence of these noncoding *PCSK9* variants on LDL-C levels.

### 3.3. Mendelian Randomization for LDL-C and DM

Previous studies including GWAS and MR analyses have demonstrated that a genetically induced elevation of circulated LDL-C levels is associated with a lower risk of DM [12,46], supporting the observation that lowering LDL-C levels with statins slightly increases the risk of DM [13,14,15,16]. However, the mechanism underlying these effects is still unknown. MR studies using LDL-C-lowering drugs’ target genes—such as *HMGCR*, *PCSK9*, and *NPC1L1* variants—as genetic instruments have consistently demonstrated an inverse association between LDL-C and DM [10,18,19,20]. However, not all MR and genetic studies have revealed inverse associations between each LDL-C-level-lowering allele and an increased risk of DM [11,21,47]. Accordingly, the heterogeneity between genetic and MR studies suggests that the diabetogenic effect of LDL-C reduction is mechanism-specific, and may depend on the underlying reduction in LDL-C levels, such as pancreatic *ß*-cell dysfunction caused by increased intracellular cholesterol levels due to either increased LDLR expression or transmembrane cholesterol transport [17,48]. Furthermore, genetic studies may shed light on the effects of pleiotropy on the association between LDL-C-level-determining alleles and DM status. For example, some studies have revealed that LDL-C-lowering alleles of *HMGCR* and *PCSK9* variants increase body weight and waist circumference [17,18,19]. *GCKR*, *TM6SF2*, and *PNPLA3* variants were reported to be associated with diabetogenic traits such as high liver fat, in addition to being associated with reductions in LDL-C levels [49]. However, most of the reported MR studies have included only people of European descent rather than people of Asian descent. Our results also reveal an inverse association between genetically determined LDL-C levels and the prevalence of DM, and the corresponding effect size is similar to that reported in previous studies. According to a WGRS derived using *PCSK9*, we noted that an LDL-C reduction of 1 mmol/L was associated with an increased risk of DM (OR 1.61, 95% CI: 1.04–2.49), and that an LDL-C reduction of 1 mg/dL was associated with an increased risk of DM (OR 1.01, 95% CI: 1.00–1.02). We obtained a similar estimate (OR 1.32: 1.09–1.60 per 1 mmol/L) when we applied a WGRS derived using LDL-C GWAS-associated variants. *PCSK9* mutations can increase or reduce LDL-C levels in the blood. If only a single lead SNV rs10788994 in the *PCSK9* gene region was used as genetic instrument for reducing LDL-C, such as *HMGCR* rs3064191 alone, the effect size would be insufficient to increase the risk of diabetes. When we applied a WGRS derived by aggregating several common and rare *PCSK9* variants, we noted that the diabetogenic effect of LDL-C reduction became evident. This finding was also supported by the association of the WGRS derived by aggregating 41 LDL-C GWAS-significant variants with DM status. Moreover, *PCSK9* variants—both common and rare—were not linked to any other metabolic traits. Overall, our findings indicate that lower circulating LDL-C levels constitute a key mediator of increased risk of DM. The diabetogenic effect can be attributed to reduction in LDL-C levels, and the association of individual SNVs with the risk of DM may be proportional to its association with LDL-C reduction.

### 3.4. Limitations

The present study has some limitations. Because of the cross-sectional study design, we could not investigate the association of circulating LDL-C at baseline with the incidence of DM; thus, we could not avoid survival bias. Second, genetic association studies revealed ethnic genetic heterogeneity; therefore, our findings may not be applicable to other ethnic groups. Third, our study did not have a second cohort; the inclusion of a second cohort—particularly one with a larger sample size and longitudinal follow-up—would strengthen the validity of our findings.

## 4. Methods and Materials

### 4.1. TWB Participants

The TWB population-based cohort study recruited participants from centers across Taiwan between 2008 and 2020. A total of 107,494 participants who were genotyped using the Axiom Genome-Wide CHB 1 or 2 Array and had no history of cancer were enrolled. Of these participants, 32,053 were excluded from the analysis because 12,289 participants did not have imputation data; 10,956 participants exhibited second-degree relatedness, as indicated by an identity with descent PI_HAT of >0.187, which was determined from a QC analysis conducted in a GWAS; 2862 participants fasted for less than 6 h; and 5946 participants had a history of hyperlipidemia. Moreover, participants with a history of DM, hypertension, hyperlipidemia, or gout—as determined from an examination of glucose metabolism parameters, blood pressure status, lipid profiles, and serum uric acid levels, respectively—were excluded. Figure 1 illustrates the flowchart of participant enrollment. The definitions of DM, hypertension, hyperlipidemia, and current smoking status are provided in Appendix A. Ethical approval was received from the Research Ethics Committee of Taipei Tzu Chi Hospital, Buddhist Tzu Chi Medical Foundation (approval number: 08-XD-005), and Ethics and Governance Council of the Taiwan Biobank (approval number: TWBR10908-01). Each participant provided written informed consent.

### 4.2. Genomic DNA Extraction and Genotyping

After blood samples were collected, DNA was extracted using a PerkinElmer Chemagic 360 instrument, in accordance with the manufacturer’s instructions (PerkinElmer, Waltham, MA, USA). Custom TWB chips were used to execute single-nucleotide variation genotyping on an Axiom Genome-Wide Array Plate system (Affymetrix, Santa Clara, CA, USA).

### 4.3. Clinical Phenotypes and Laboratory Examinations

The following clinical phenotypes were used in this study: BMI; waist circumference; waist–hip ratio; and systolic, mean, and diastolic blood pressure. Additionally, the following biochemical and hematological parameters were used: glucose metabolism parameters such as hemoglobin A1c (HbA1c) and fasting plasma glucose levels; liver and renal function parameters such as serum creatinine, blood urea nitrogen, aspartate aminotransferase (AST), alanine aminotransferase (ALT), γ-glutamyl transferase (γ-GT), albumin, total bilirubin, and uric acid levels; lipid profiles such as total cholesterol, high-density lipoprotein cholesterol (HDL-C), and LDL-C and triglyceride levels; and platelet counts, white and red blood cell counts, and hemoglobin and hematocrit levels. The BMI and estimated glomerular filtration rate (eGFR) were calculated as reported previously [50]. Because of the absence of data related to urine creatinine levels, only spot urine albumin levels were used to evaluate albuminuria.

### 4.4. Regional Plot Analysis and GWAS

A regional plot association analysis was conducted with LocusZoom.js [51] to determine the lead SNVs providing genome-wide significant evidence of association within the *PCSK9* region for LDL-C levels. First, we performed a QC analysis in the GWAS to assess the participants who were enrolled after the application of the exclusion criteria (Figure 1). In the analysis, we used Axiom Genome-Wide CHB 1 and 2 Array plates (Affymetrix, Santa Clara, CA, USA) with 24,927 and 69,529 participants and comprising 611,656 and 640,160 SNVs, respectively, for genome-wide genotype imputation. The reference panel for this imputation process was the East Asian population from the 1000 Genome Project Phase 3 study, and the imputation tools were SHAPEIT and IMPUTE2. After the imputation process, the QC analysis was performed by filtering SNVs with an IMPUTE2 imputation quality score of >0.3. Indels were removed using VCF tools. All samples included in the analysis had a call rate of ≥97%. In the SNV QC analysis, SNVs with a call rate of <97%, a minor allele frequency of <0.01, and a violation of the Hardy–Weinberg equilibrium (*p* < 10^−6^) were excluded from subsequent analyses. Thus, after the QC analysis, we included 75,441 participants as well as 330 SNVs located at positions between 55.40 and 55.65 Mb on chromosome 1p32.3 for the regional plot association analysis. Subsequently, we extended the analysis to execute a whole-genome genotyping and imputation. After the execution of the QC analysis and application of other exclusion criteria, the same 75,441 participants and 3,639,888 SNVs were included for the GWAS of LDL-C levels. A *p*-value of <5 × 10^−8^ was considered to be a genome-wide significant association.

### 4.5. Statistical Analysis

Continuous variables are presented herein as means ± standard deviations or medians and interquartile ranges. The Mann–Whitney U test was used to compare differences in the distribution of linear parameters between the sexes. Lipid profiles and urine albumin levels were transformed logarithmically when examined using an analysis of variance and regression. Categorical data are presented herein as percentages, and their distributions were compared using the chi-squared test. Under the assumption of an additive genetic effect, we used a general linear regression model to analyze the correlation between the studied phenotypes, genotypes, and confounders after adjustment for age, sex, BMI, and current smoking status. We also applied a multiple logistic regression analysis to evaluate the independent effects of the investigated genotypes on lifestyle risks and atherosclerotic risk factors. Moreover, a linear stepwise regression analysis was conducted to determine the independent correlates of LDL-C levels. Genome-wide scans were performed using the PLINK software package. A *p*-value of <5 × 10^−8^ was considered to indicate a genome-wide significant association. In our genotype–phenotype analysis, we applied the Bonferroni correction. Statistical significance was defined as *p* < 0.0015, which was calculated as 0.05/33 according to 33 traits analyzed. LD was calculated using LDmatrix software (https://analysistools.nci.nih.gov/LDlink/?tab=ldmatrix (accessed on on 25 May 2022)). All statistical analyses were performed using SPSS (version 22; SPSS, Chicago, IL, USA).

### 4.6. MR Analysis

2SLS regression with IVs was performed to examine whether the LDL-C-determining SNVs and WGRSs were associated with the risk of DM through their associations with LDL-C levels. During the first stage of the regression, LDL-C-determining SNVs and WGRSs were regressed to generate predicted LDL-C levels. During the second stage of the regression, the study parameters were regressed on LDL-C-determining SNVs and WGRSs to generate the predicted risk of DM. To create WGRSs, SNVs were weighted according to each allele score by using the *β* coefficients from our GWAS analysis, and the risk allele exhibiting directionally concordant associations with the target parameters was selected. The F statistic was used to assess the strength of the instruments, and was calculated using the following equation: F = R^2^(n − 2)/(1 − R^2^), where R^2^ is the proportion of the variability in genetically determined LDL-C levels accounted for by the SNVs, and n is the sample size [52]. An F statistic of >10 indicates a relatively low risk of weak instrumental bias in MR analyses [52].

### 4.7. Scatterplots and Testing for Heterogeneity

The concordance of IV effects in the generalized linear models was tested using scatterplots [53]. In general, a causal conclusion may seem reasonable if several independent genetic variants in different gene regions are concordantly associated with the outcome. In this study, heterogeneity was assessed either visually in the scatterplot or through Cochran’s Q test on the causal estimate from each genetic variant with the outcome against the genetic association of the exposure of interest, in addition to their confidence intervals [54,55].

### 4.8. Sensitivity Analysis

Details of the sensitivity analysis are provided in Appendix A, including a multivariate analysis, funnel plots, IVW methods, simple and weighted median methods, and MR–Egger regression, which were reported previously [50].

## 5. Conclusions

Common and rare *PSCK9* variants are independently associated with LDL-C levels in the Taiwanese population. The results of MR analyses executed using genetic instruments based on WGRSs derived from *PCSK9* variants or LDL-C GWAS-associated variants demonstrate a causally inverse association between LDL-C levels and DM. These results provide further evidence that *PCSK9* variants and other LDL-C GWAS-associated variants may play a crucial role in determining cardiometabolic outcomes.

## Figures and Tables

**Figure 1 ijms-23-10418-f001:**
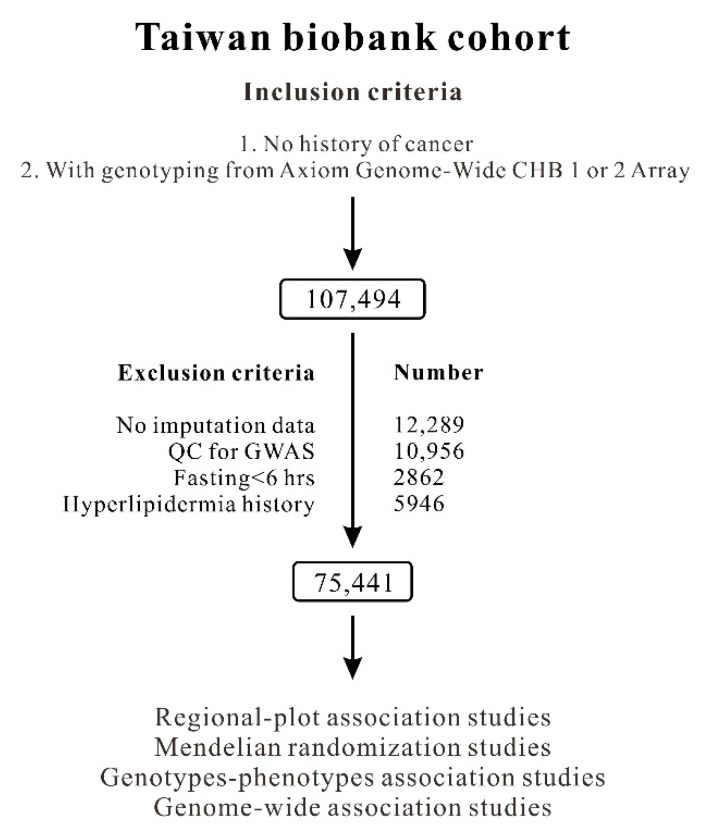
Participant selection flowchart.

**Figure 2 ijms-23-10418-f002:**
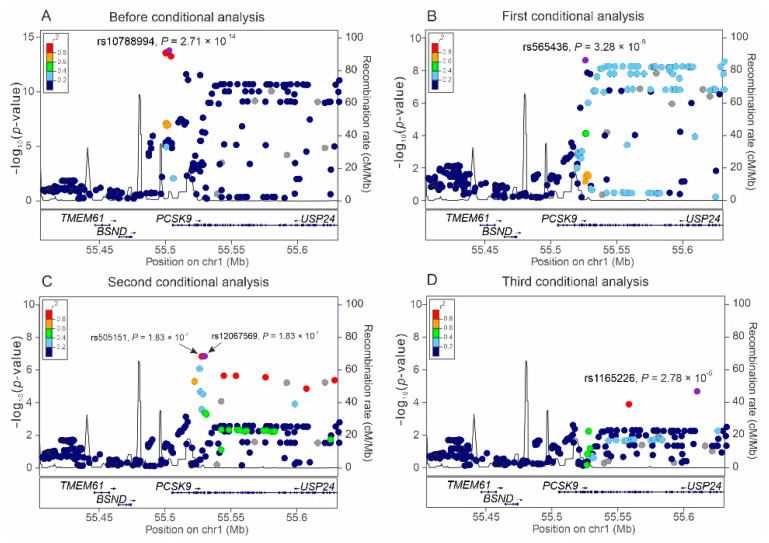
Regional association plots in a region of 200 kb surrounding the *PCSK9* locus on chromosome 1p32 for low-density lipoprotein cholesterol levels. Regional plot associations are shown without (**A**) or with serial conditional analysis after further adjustment for rs10788994 (**B**), rs565436 (**C**), and rs505151 (**D**) genotypes.

**Figure 3 ijms-23-10418-f003:**
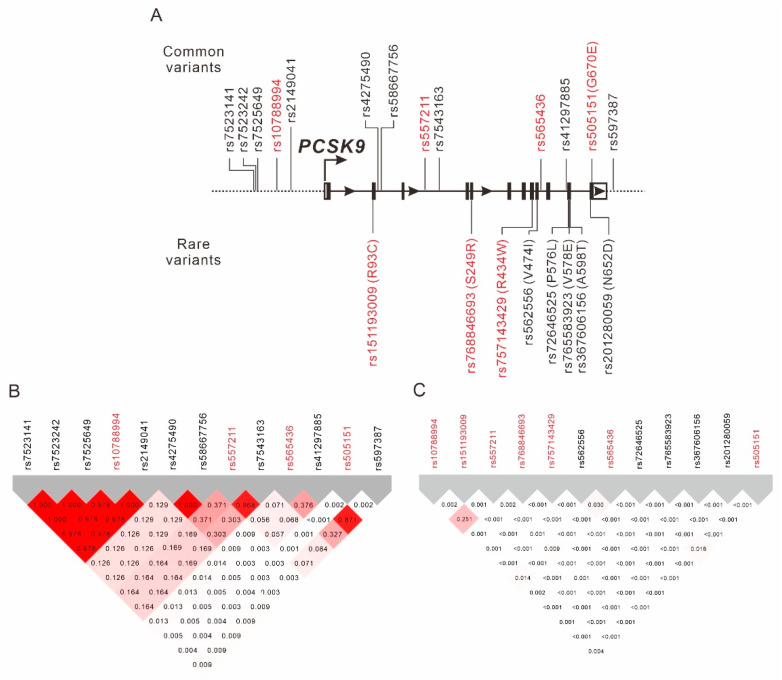
Location (**A**) and linkage disequilibrium (**B**,**C**) of common and rare *PCSK9* variants. Shades of red and gray show the strength of the pairwise linkage disequilibrium based on r^2^, and numbers indicate the value of r^2^. Seven *PCSK9* variants associated with LDL-C levels for weighted genetic risk scores are highlighted in red font.

**Table 1 ijms-23-10418-t001:** Association between *PCSK9* variants of low-density lipoprotein cholesterol (LDL-C) levels from Axiom Genome-Wide CHB 1 and 2 Array plates with genotype imputation, including common and rare *PCSK9* variants.

*PCSK9* Variants	Chr	Position	Ref/Alt	Func.refGene	Gene.refGene	HWE	MAF	MM	Mm	mm	*p* Value	Beta	SE	*p* * Value
rs10788994	1	55,500,976	C/T	intergenic	*BSND;PCSK9*	0.7616	0.3475	121.80 ± 31.26 (31,728)	120.41 ± 30.84 (33,743)	119.23 ± 30.86 (9015)	6.44 × 10^−13^	−0.0047	0.0006	1.99 × 10^−14^
rs151193009	1	55,509,585	C/T	exon2:c.C277T:p.R93C	*PCSK9*	0.4419	0.0027	120.98 ± 30.98 (73,449)	106.76 ± 28.53 (405)	--	1.76 × 10^−18^	−0.0579	0.0057	1.19 × 10^−24^
rs557211	1	55,514,215	T/G	Intron Variant	*PCSK9*	0.8433	0.1916	121.42 ± 31.179 (49,044)	119.81 ± 30.701 (23,300)	119.30 ± 30.654 (2740)	1.69 × 10^−11^	−0.0052	0.0007	3.75 × 10^−12^
rs768846693	1	55,518,412	C/A	exon5:c.C747A:p.S249R	*PCSK9*	0.9078	0.0004	120.88 ± 31.02 (75,324)	91.73 ± 28.18 (63)	--	1.18 × 10^−9^	−0.1301	0.0143	1.12 × 10^−19^
rs757143429	1	55,523,828	C/T	exon8:c.C1300T:p.R434W	*PCSK9*	0.7454	0.0011	120.89 ± 31.02 (75,239)	107.81 ± 30.25 (175)	--	1.47 × 10^−7^	−0.0509	0.0086	3.46 × 10^−9^
rs565436	1	55,524,601	G/A	Intron Variant	*PCSK9*	0.3577	0.1033	121.22 ± 31.11 (60,123)	119.41 ±30.65 (13,901)	118.19 ± 31.05 (780)	8.08 × 10^−10^	−0.0067	0.0010	3.81 × 10^−12^
rs505151	1	55529187	G/A	exon12:c.G2009A:p.G670E	*PCSK9*	0.9202	0.0534	119.0 ± 31.00 (67,530)	120.00 ± 31.21 (7603)	121.00 ± 32.37 (215)	6.00 × 10^−6^	0.0065	0.0013	5.89 × 10^−7^

*p*: unadjusted; *p* *: adjusted for age, sex, BMI, and current smoking status. Abbreviations: SNV, single-nucleotide variation; Chr: chromosome; Ref, reference allele; Alt, alternate allele; HWE, Hardy–Weinberg equilibrium; MAF, minor allele frequency; MM: homozygosity of major allele; Mm: heterozygosity of major and minor alleles; mm: homozygosity of minor allele; SE, standard error. Data are presented as the mean ± standard deviation (mg/dL) (number). LDL cholesterol values were logarithmically transformed before statistical testing to produce a normal distribution; however, the untransformed data are shown.

**Table 2 ijms-23-10418-t002:** Serum total and low-density lipoprotein cholesterol (LDL-C) levels: stepwise linear regression analysis, including *PCSK9* genotypes.

	Serum Total Cholesterol Level			Serum LDL-C Level		
	Beta	r^2^	*p* Value	Beta	r^2^	*p* Value
Age (years)	0.0014	0.0362	<10^−307^	0.0015	0.0186	<10^−307^
Sex (male vs. female)	0.0154	0.0049	1.15 × 10^−131^	--	--	--
Body mass index (kg/m^2^)	0.0017	0.0066	1.09 × 10^−108^	0.0051	0.0268	<10^−307^
Current smoking status (%)	0.0060	0.0004	8.49 × 10^−9^	--	--	--
rs10788994 (TT vs.TC vs. CC)	−0.0022	0.0007	9.21 × 10^−6^	−0.0039	0.0009	6.80 × 10^−8^
rs151193009 (CC vs. CT)	−0.0388	0.0013	1.47 × 10^−24^	−0.0615	0.0014	3.03 × 10^−27^
rs557211 (TT vs.TG vs. GG)	−0.0018	0.0001	0.0024	−0.0024	0.0001	0.0052
rs768846693 (CC vs. CA)	−0.0662	0.0006	8.4 × 10^−12^	−0.1261	0.0011	3.87 × 10^−18^
rs757143429 (CC vs. CT)	−0.0310	0.0004	7.69 × 10^−8^	−0.0511	0.0005	3.43 × 10^−9^
rs565436 (AA vs. AG vs. GG)	−0.0038	0.0005	7.55 × 10^−9^	−0.0060	0.0005	1.19 × 10^−9^
rs505151 (AA vs. AG vs. GG)	0.0036	0.0002	0.0001	0.0065	0.0003	1.23 × 10^−6^

**Table 3 ijms-23-10418-t003:** Summary of coefficients used for standard Mendelian randomization analysis: low-density lipoprotein cholesterol (LDL-C) levels and diabetes mellitus (DM).

T_A_	T_B_	G_A_	T_A_-T_B_			G_A_-T_A_			G_A_-T_B_			IV_A_-T_B_		
			Beta	SE	*p ^a^*	Beta	SE	*p ^a^*	Beta	SE	*p ^a^*	Beta	SE	*p ^a^* (*p ^b^*)
LDL-C	DM	WGRS_*PCSK9*_7SNVs	−1.9993	0.1185	6.76 × 10^−64^	0.5599	0.0301	4.66 × 10^−77^	−2.3685	0.8918	0.0079	−4.2294	1.5926	0.0079 (0.0098 *^c^*)
		WGRS_LDL-C_41SNVs	−1.9993	0.1185	6.76 × 10^−64^	0.9823	0.0202	<10^−307^	−1.9743	0.6972	0.0046	−1.9710	0.6961	0.0046 (5.02 × 10^−7^ *^c^*)

WGRS_*PCSK9*_7SNVs: weighted genetic risk scores (WGRSs) of seven *PCSK9* variants; WGRS_LDL-C_41SNVs: WGRSs of 41 LDL-C genome-wide association study (GWAS)-significant variants; IVA and IVB: instrumental variables for GA and GB, respectively. *a*: Adjustment for age, sex, current smoking status, and BMI. *b*: After further adjustment of LDL-C levels, the *p*-values were 0.1256 for WGRS_*PCSK9*_7SNVs and 0.7606 for WGRS_LDL-C_41SNVs. *c*: Adjustment for age, sex, current smoking status, BMI, and other possible confounders, such as eGFR, platelet counts, hemoglobin, triglyceride, and AST.

## Data Availability

The data presented in this study are available upon request from the corresponding author.

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
