# Peer review of "Common and Rare PCSK9 Variants Associated with Low-Density Lipoprotein Cholesterol Levels and the Risk of Diabetes Mellitus: A Mendelian Randomization Study"

_ijms, 2022, doi:10.3390/ijms231810418_

Round 1

Reviewer 1 Report

Common and rare PCSK9 Variants

The Taiwan Biobank is a remarkable data base and has spawned many original and interesting papers. The present report examines PCSK 9 variants in the study population and finds new variants associated with serum cholesterol levels and some with prevalent diabetes. Interestingly but peehaps not surprising there was an association between LDL cholesterol and diabetes but the association became non significant after adjustment for LDL-Cholesterol levels.This should be added to the summary for clarification although I accept that hypercholesterolaemia may be associated with B cell dysfunction due to oxidative stress

In conclusion and interesting report with new information based on the Taiwanese databank

To the Editor

Another trawling exercise of the large Taiwanese population. Worthy of publication

Reviewer 2 Report

This paper by Lung-An Hsu and colleagues aims to identify the role of PASK9 and genome-wide variants associated with LDL-C levels and explore the causal relationship between LDL-C and diabetes mellitus. They performed a regional and genome-wide association study using individuals from the Taiwan Biobank dataset and identified 7 common or rare variants in the PCSK9 locus and 41 GWAS variants associated with LDL-C levels. MR analyses demonstrated an inverse association between LDL-C levels and DM. The manuscript is very well organized and written with a high degree of clarity. But I still have some questions.

1.     In the introduction part, the authors need to explain why do they focused on PCSK9 instead of other genes that also play critical roles in LDL-C?

2.     In Supplementary Table 1, the significance seems incorrect. For example, the median age of male participants and female participants are the same, it is hard to believe that the p < 0.0001 is correct.

3.     The figure legends should be improved. For example, what does the red font mean in figure 3?

4.     The authors used r2<0.3 (line 105) to select independent SNPs significantly associated with LDL-C in PCSK9 regions, which is not usually used in MR analyses (r2<0.001).

5.     To meet the assumption of MR, only the genetic variants that are strongly associated with the exposure should be included. Why did the authors include rs505151 (1.83×10-7) which did not reach the genome-wide significance threshold (P < 5×10-8) in the analyses?

6.     Line 120, how the eight rare exonic nonsynonymous mutations were selected?

7.     Line 152, the authors calculated the association between LDL-C levels and DM prevalence. Considering that participants with a history of DM were included in the TWB (line 342), how did the analysis was performed?

8.     What’s the definition of lead SNVs (line 160) and how to get these lead SNVs?

Round 2

Reviewer 2 Report

I don't have further comments.